# Herpes simplex virus 1 strain 17+ with R2 mutation in UL37 has residual retrograde transport

Marius Walter,[1] Anoria K. Haick,[1] Paola A. Massa,[1] Lindsay M. Klouser,[2] Laurence Stensland,[2] Tracy K. Santo,[2] Hong Xie,[2] Keith R. Jerome[1,2]

**ABSTRACT** Herpes simplex virus 1 (HSV-1) causes lifelong recurrent infections. Following primary infection of the oral or genital mucosa, HSV-1 travels retrogradely through axons and establishes latency in the cell body of ganglionic neurons of the peripheral nervous system. Periodic reactivation in neurons and anterograde transport of virions back to peripheral regions cause oral or genital ulcerations. Many host and viral factors implicated in retrograde and anterograde transport of HSV-1 have been identified. In particular, studies reported that introducing five amino acid substitutions in the R2 region of the viral tegument protein UL37 was sufficient to completely eliminate retrograde transport of HSV-1 strain F. Here, we introduced the same R2 mutations in the highly neurovirulent HSV-1 strain 17+. We show that this $R2^{17}$ virus is highly attenuated in mice and acts as a potent vaccine that protects mice against acute HSV-1 infection. However, we report that the $R2^{17}$ virus has residual retrograde transport. We show that $R2^{17}$ can establish latency in mouse models of ocular and vaginal infection and reactivate. These results contradict published evidence and show that the R2 mutation is not sufficient to fully prevent retrograde transport of HSV-1.

**IMPORTANCE** Herpes simplex virus 1 (HSV-1) is a ubiquitous pathogen without a cure or vaccine. HSV-1 travels through nerves between the oral and genital mucosa and the peripheral nervous system, where it establishes lifelong latency. Studies reported that introducing five amino acid substitutions in the R2 region of the viral tegument protein UL37 was sufficient to completely eliminate the retrograde transport of HSV-1 strain F from the mucosa to the nervous system. Here, we present contradictory findings. We report that an HSV-1 virus from strain 17+ with the same R2 mutation has residual retrograde transport. This shows that the R2 mutation is not sufficient to fully prevent the retrograde transport of HSV-1 in all settings. This finding may be particularly relevant for assessing the safety of prospective live-attenuated vaccines that include the R2 mutation.

**KEYWORDS** herpes simplex virus, retrograde transport, virology

Herpes simplex virus 1 (HSV-1) persistently infects close to 70% of the human population and causes oral and genital ulceration. After initial infection of epithelial tissues of the oral or genital mucosa, virions enter axons and are transported retrogradely to the cell bodies of sensory or autonomic neurons in peripheral ganglia, where the viral DNA genome remains latent for life. Periodic reactivation of the latent genome and anterograde transport of virions back to the mucosal periphery cause viral shedding and ulcers.

Numerous studies have identified the mechanisms implicated in retrograde and anterograde transport of HSV virions, as reviewed in references 1–3. After viral entry in the axon tip, unenveloped capsids attach to dynein motors and are transported

Address correspondence to Marius Walter, mwalter2@fredhutch.org.

The authors declare no conflict of interest.

See the funding table on p. 8.

retrogradely to neuronal cell bodies. Studies have shown that retrograde transport requires the viral inner tegument proteins UL36 and UL37. UL36 directly binds to the viral capsid and attaches to the dynein/dynactin microtubule motor complex, promoting the trafficking of the capsid toward the nucleus (4). UL37 is a protein deamidase that binds to UL36 and is also necessary for retrograde transport (5–8). The UL37 N-terminus contains several domains evolutionarily conserved among related alphaherpesviruses, named R1, R2, and R3 (7, 8). Studies have shown that introducing five alanine substitutions in the R2 domain did not affect the overall protein structure of UL37 but rendered the virus avirulent (8). These defined substitutions, referred hereafter as the R2 mutation, were introduced in HSV-1 strain F as well as in the related pseudorabies virus (PRV) and bovine herpesvirus 1 (BoHV-1) (8–11). In all cases, the R2 mutation appeared to completely eliminate retrograde transport. In particular, the HSV-1 R2$^F$ virus replicated in the periphery but could not be detected in the trigeminal ganglia and dorsal root ganglia following ocular infection in mice and genital infection in guinea pigs, respectively. Similarly, the PRV R2 virus was not detected in the trigeminal ganglia after intranasal infection in mice or pigs, and the BoHV-1 R2 virus was not detected in the trigeminal ganglia after intranasal or ocular infection in calves (8–11). Importantly, these studies demonstrated that R2 viruses acted as safe live-attenuated vaccines, protecting against subsequent challenge with wild-type HSV or PRV.

For unrelated vaccine studies, we wished to use a virus with no retrograde transport and compare it with other live-attenuated vaccine candidates that we are developing. We introduced the R2 mutation into HSV-1 strain 17+, a strain highly neurovirulent in mice, from which all our engineered viruses are derived. While testing this R2$^{17}$ virus side by side with other vaccine candidates, we noticed that R2$^{17}$ could establish latency in the ganglia after ocular or vaginal infection of mice. These observations show that R2$^{17}$ has residual retrograde transport, which contradicts the published studies mentioned above (8–10). In this manuscript, we report these findings as a standalone observation. They will be of interest to the field and show that the R2 mutation is not sufficient to fully prevent retrograde transport of HSV-1 in all settings.

## RESULTS AND DISCUSSION

We aimed to build an HSV-1 virus that would replicate efficiently in epithelial tissues but did not invade the nervous system. We thus incorporated the five R2 alanine substitutions (UL37-Q403A, E452A, Q455A, Q511A, R515A) into the genome of the highly neurovirulent HSV-1 strain 17+. The mutations were added by CRISPR-mediated homologous recombination of a synthesized gene fragment containing the five mutations. As a recipient virus, we used an HSV-1 strain 17+ isolate that also carried a cyan fluorescent reporter (CFP) inserted between the *US1* and *US2* viral genes (12). Viral clones were plaque-purified and the presence of the R2 mutation was confirmed by Sanger sequencing (Fig. 1A). Shotgun sequencing of the viral stock showed that R2$^{17}$ included the UL37 mutations with no trace of wild-type sequences (Table S1). In addition, the R2$^{17}$ genome included 24 polymorphisms compared with the reference HSV1–17+ genome, with seven missense mutations of unknown effect in other protein-coding genes.

To use as a challenge virus from a different strain for vaccination studies, we also built a McKrae-YFP virus by adding a YFP reporter to the HSV-1 strain McKrae. Surprisingly, shotgun sequencing of this new virus revealed that it was a recombinant between McKrae and 17+ strains, with regions of the new virus originating from either genome (Fig. 1B). We traced the mistake to a contamination in the McKrae stock received from a collaborator. For the sake of clarity, we refer to this YFP-expressing McKrae/17+ recombinant as McKrae-17 in the following paragraphs. Both R2$^{17}$ and McKrae-17 replicated well in cell culture, with no noticeable differences from their parental strains.

Our initial goal was to conduct vaccination studies and to compare R2$^{17}$ with other vaccine candidates. However, R2$^{17}$ did not behave as anticipated and did not represent the control treatment that we were expecting. In the following paragraphs, we present

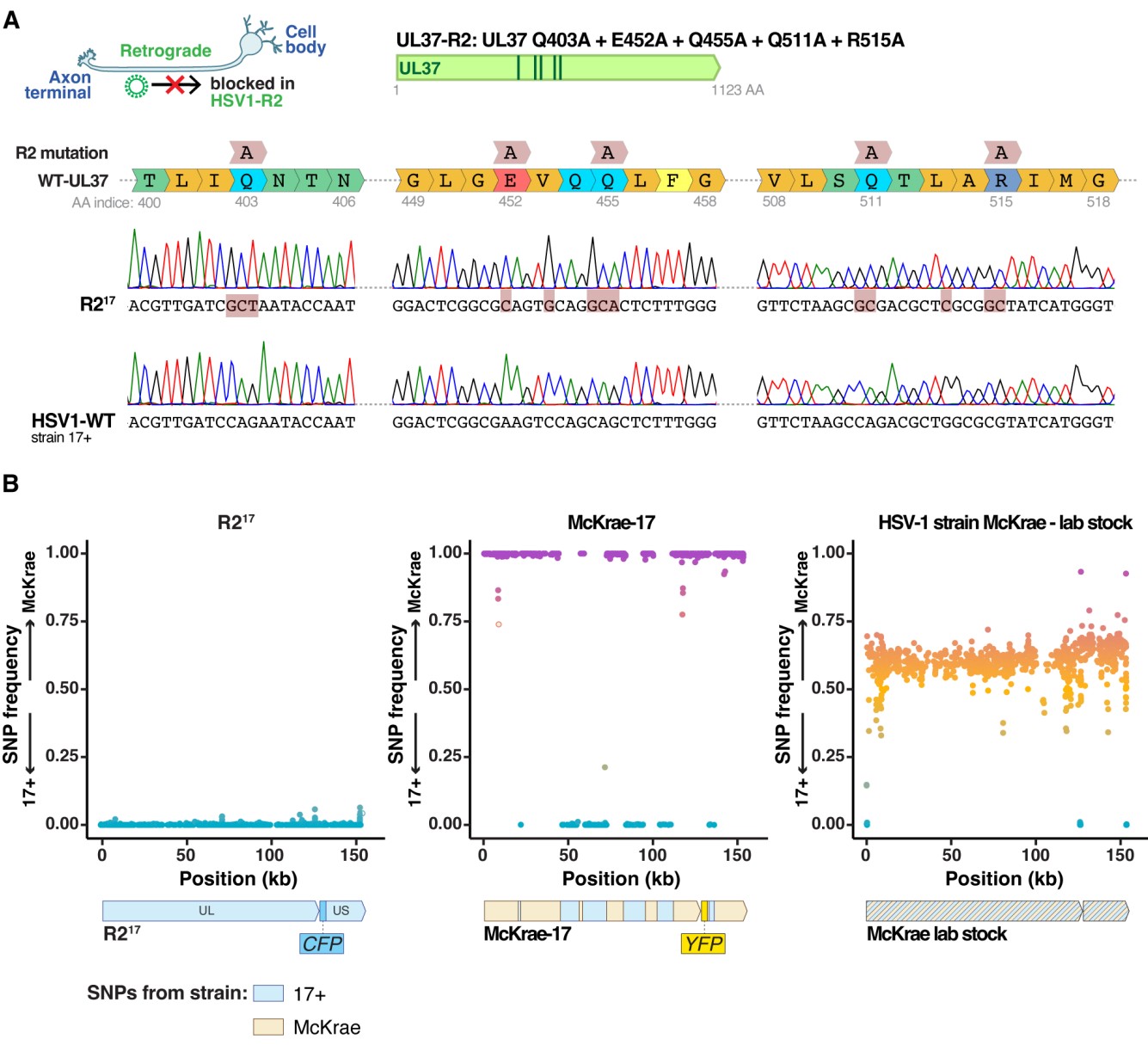

**FIG 1** Characterization of R2[17] and McKrae-17 virus. (A) R2[17] was generated by introducing five alanine substitutions in the inner tegument protein UL37: Q403A + E452A + Q455A + Q511A + R515A. The figure shows Sanger sequencing chromatogram of the coding sequence of the *UL37* gene of R2[17] and wild-type HSV-1 strain 17+. (B) Fraction of SNPs of strain 17+ or McKrae origin alongside the HSV-1 genome after shotgun sequencing of R2[17], McKrae-17, and the McKrae stock obtained from a collaborator. Each dot represents an individual SNP. McKrae-17 was a recombinant virus between strain 17+ and McKrae (middle panel).

the results observed in the R2[17] arm of these studies as a standalone observation, since these contradictory findings could be of interest to researchers in the field. The rest of our vaccination studies will be presented elsewhere.

First, we tested if R2[17] could be used as a preventive vaccine to protect against HSV-1 ocular infection (Fig. 2A). Female Swiss Webster mice were inoculated ocularly after corneal scarification with R2[17], at $10^6$ plaque-forming units (PFU) per eye. In this model, infection with $10^5$ PFU of wild-type HSV-1 strain 17+ typically causes extensive facial lesions and mortality before establishing latency in the trigeminal ganglia (TG) and other nervous areas (13–15). However, infection with R2[17] did not cause visible symptoms, and no mortality was observed. Four weeks after vaccination, mice were challenged ocularly with a high dose of McKrae-17 ($10^7$ PFU per eye). Unlike unvaccinated mice, vaccinated animals survived the challenge without symptoms (Fig. 2B). Together, these

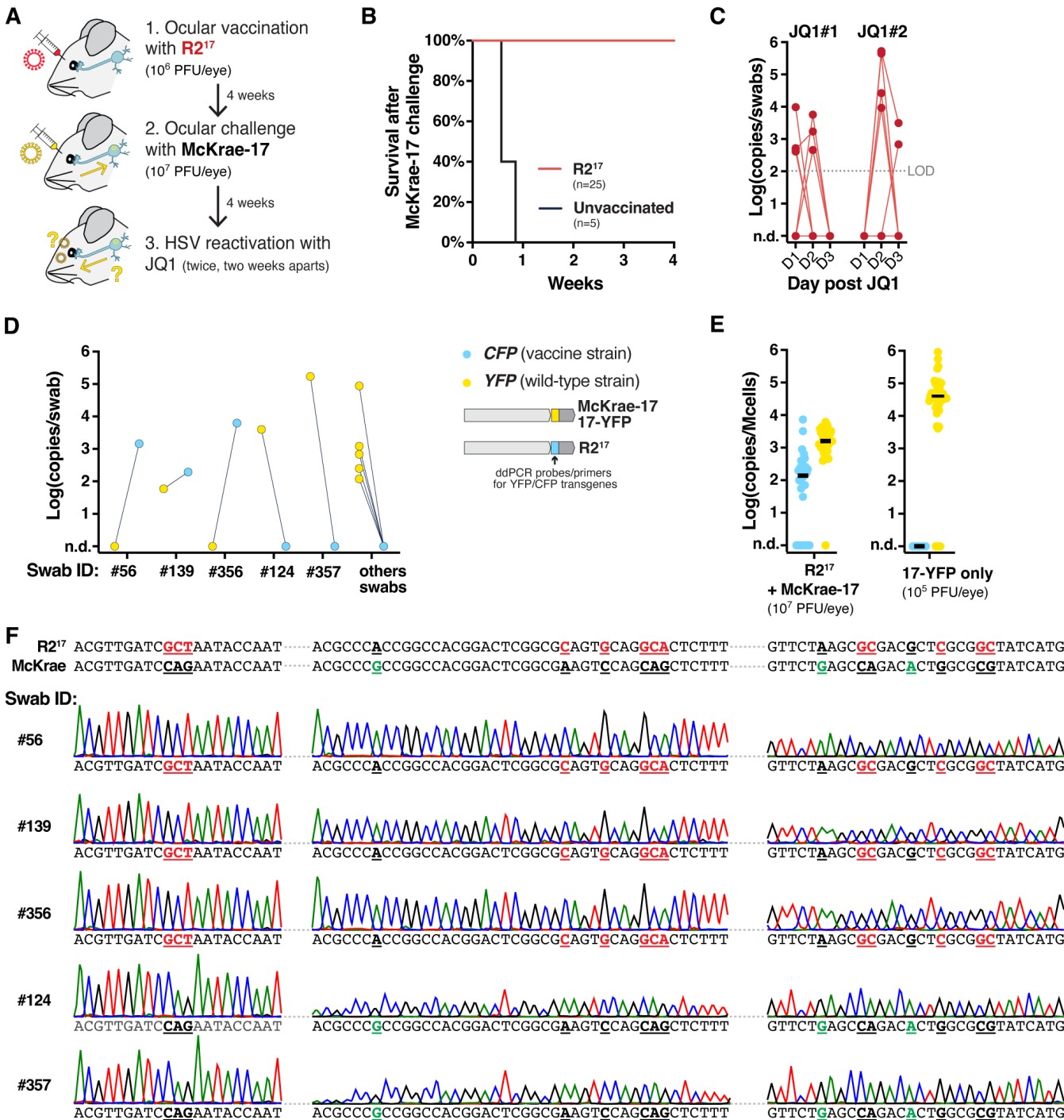

FIG 2 R2[17] establishes latency and can reactivate after ocular infection. (A) Experimental design. (B) Survival of vaccinated mice after challenge with McKrae-17, $n$ = 25 (R2[17] vaccination), or $n$ = 5 (unvaccinated). (C) Viral shedding after JQ1 reactivation. Data show viral titers measured by qPCR in eye swabs at days 1–3 after JQ1 treatment. Consecutive datapoints from individual mice are connected by a line. $n$ = 25 mice. Titers are expressed in log-transformed copies per swab. n.d.: non-detected. LOD: limit of detection. (D) Genotyping of positive swabs by duplex ddPCR, detecting the CFP and YFP markers present in R2[17] and McKrae-17 virus, respectively. (E) Latent viral load in the TG by duplex ddPCR after R2[17] vaccination and McKrae-17 challenge (Left). The right panel shows the latent viral load after ocular infection with an HSV-1 strain 17+ expressing YFP (17-YFP) at $10^5$ PFU/eye. Titers are expressed in log-transformed copies per million cells after normalization with mouse *RPP30* levels. Black lines indicate the median. $n$ = 15 mice, with two TG per mouse. (F) Sanger sequencing of the R2 region of swabs expressing *CFP* (Swabs #56, 139, 356) or *YFP* (#124, 357). Bases highlighted in red indicate bases mutated in R2[17]. Bases highlighted in green indicate SNPs between the 17+ and McKrae strains.

results aligned well with published studies. It showed that R2[17] was avirulent and acted as a potent vaccine that protected against HSV-1 acute infection.

Next, we tested if vaccination had protected against latency establishment and viral shedding. HSV-1 usually does not reactivate spontaneously in mice, but the virus can be artificially reactivated and detected in eye swabs by treating animals with the bromodomain inhibitor JQ1 (13–15). HSV-1 shedding is typically detected in 10%–50% of mice using this method. Four weeks after McKrae-17 challenge, mice were treated twice with JQ1, 2 weeks apart, and levels of reactivated viruses were quantified in eye swabs by qPCR (Fig. 2C). HSV-1 shedding was detected in 20%–25% of vaccinated mice. The reactivated swabs were genotyped using a duplex droplet digital (dd)PCR assay that distinguished between the CFP and YFP reporters present in the vaccine and challenge strains, respectively (Fig. 2D). Surprisingly, the CFP reporter present in R2[17] was detected in three swabs originating from three different mice (swab ID #56, #139 and #356). Sanger sequencing of the R2 region of these reactivated swabs showed that they carried the R2 mutation (Fig. 2F). By contrast, sequencing of two swabs expressing only the YFP reporter (swab ID #124 and #357) showed that they did not carry the R2 mutation but had SNPs specific to the McKrae strain, suggesting that McKrae-17 had reactivated in these mice. This unexpected result showed that R2[17] could reactivate, suggesting that it had established a latent infection.

To confirm this hypothesis, TG were collected, and the latent viral loads of the vaccine and challenge viruses were measured by duplex ddPCR (Fig. 2E). Surprisingly, both the CFP and YFP reporters were detected in the TG, indicating that both R2[17] and McKrae-17 had established a latent infection. R2[17] was detected in 22 out of 30 TG collected from 15 mice. Overall, the latent viral load of R2[17] was reduced by two orders of magnitude compared with infection of naive mice with wild-type HSV-1 strain 17+ at $10^5$ PFU/ eyes (Fig. 2E, right panel). This showed that R2[17] could travel retrogradely to the TG to establish latency, albeit at reduced levels.

To confirm these results, we tested if vaccination with R2[17] could protect against HSV-1 infection during genital infection (Fig. 3A). Female Swiss-Webster mice were inoculated vaginally with $10^5$ or $10^6$ PFU of R2[17]. Vaginal infection with as low as $10^{3\text{-}4}$ PFU of HSV-1 strain 17+ is usually highly lethal in mice, but no symptoms or mortality were observed after vaccination with R2[17], confirming that the virus was avirulent. Four weeks later, mice were challenged vaginally with a high dose of McKrae-17 ($10^7$ PFU). All mice survived without symptoms, showing that R2[17] acted as an efficient vaccine (Fig.

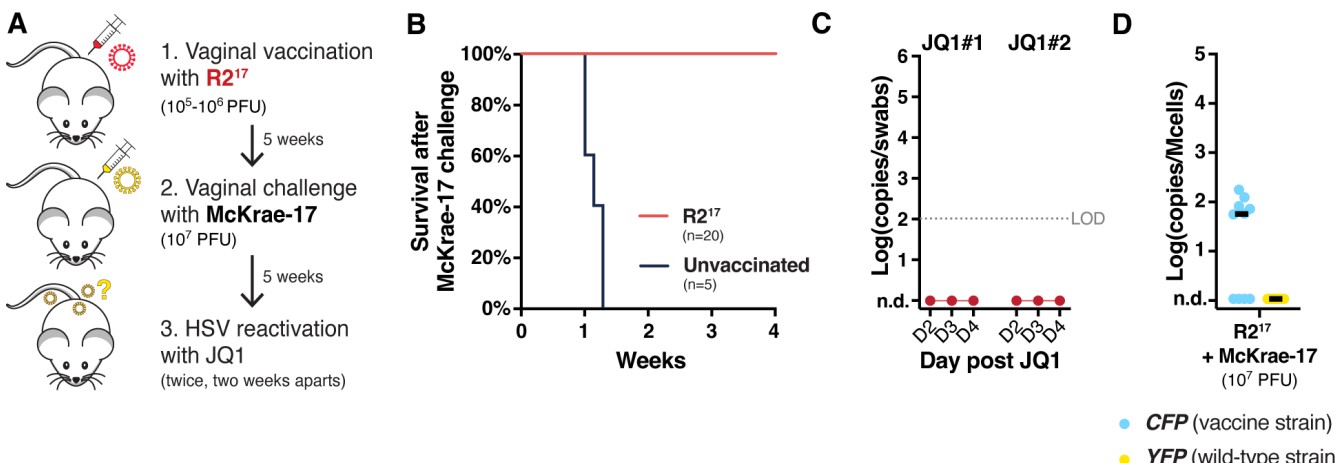

**FIG 3** R2[17] establishes latency after vaginal infection. (A) Experimental design. (B) Survival of vaccinated mice after challenge with McKrae-17, n = 20 (R2[17] vaccination), or n = 5 (unvaccinated). (C) Viral shedding after JQ1 reactivation. Data show viral titers measured by qPCR in eye swabs at days 2–4 after JQ1 treatment. n = 20 mice. Titers are expressed in log-transformed copies per swab. n.d.: non-detected. LOD: limit of detection. (D) Latent viral load in the DRG by duplex ddPCR after R2[17] vaccination and McKrae-17 challenge. Titers are expressed in log-transformed copies per million cells after normalization with mouse *RPP30* levels. Black lines indicate the median. n = 10 mice.

3B). Five weeks after challenge, mice were treated twice with JQ1, 2 weeks apart, and HSV shedding levels were measured in vaginal swabs. This time, no mice reactivated HSV-1 (Fig. 3C). Dorsal root ganglia (DRG) were collected, and the latent viral loads of the vaccine and challenge viruses were measured by duplex ddPCR (Fig. 3D). Again, we found that R2[17] had established a latent infection. R2[17] was detected at low levels in 6 out of 10 DRG, while McKrae-17 could not be detected.

Altogether, our work confirmed that HSV strains with the R2 mutation were avirulent and could act as potent live-attenuated vaccines. However, we found that R2[17] had residual retrograde transport and could establish a latent infection after ocular or genital infection in mice. R2[17] retrograde transport appeared reduced compared to wild-type HSV-1 strain 17+. These unexpected results contradict published observations that showed that the R2 mutation completely eliminated retrograde transport of HSV-1 strain F and PRV (8–10). On the contrary, our findings suggest that the R2 domain of UL37 is implicated in retrograde transport but that other viral factors can compensate for its absence.

Several hypotheses could explain these differences. HSV-1 strain F is naturally less virulent than strain 17+ and high doses are necessary to establish a potent infection (16). It is possible that the R2 mutation in a viral strain that already has low virulence, like strain F, managed to fully eliminate retrograde transport, or at least pushed it below the limit of detection. It would be of interest to compare the R2 mutants of strain F and 17+ side by side to evaluate the differences. The R2 mutation also suppressed retrograde transport of PRV and BovHV-1 (8, 11). PRV and BovHV-1 are *Varicelloviruses*, while HSV-1 belongs to the distinct genus of *Simplexviruses*, and the amino acids mutated in R2[PRV], R2[BovHV1], and R2[HSV] are only partially conserved (8). It is possible that *Varicelloviruses* and *Simplexviruses*, or at least, PRV, BovHV-1, and HSV-1, rely on slightly different viral factors for retrograde transport. Such a distinction is observed for anterograde transport, where knock-out of US9 in PRV fully eliminates anterograde transport, while US9-null mutants have residual anterograde transport with HSV-1.

## MATERIALS AND METHODS

### Cells and viruses

Viruses were propagated and engineered using African green monkey epithelial Vero cells. Vero cells were obtained from the ATCC and cultured in DMEM (Corning, Corning, NY, USA) supplemented with 10% FBS (Sigma-Aldrich, St-Louis, MO, USA). Cells were maintained at 37°C in a 5% $CO_2$ humidified incubator and frequently tested negative for mycoplasma contamination. Viral infections and plaque assays were conducted using DMEM with 2% FBS, as described previously (13).

R2[17] was generated by modifying HSV-1 strain 17+ expressing cyan fluorescent protein mTurquoise2 (HSV1-CFP), a gift from Matthew Taylor (12). A 592-bp gene fragment coding for the R2 mutation (UL37-Q403A, E452A, Q455A, Q511A, R515A) was purchased from IDT (USA) and introduced into HSV1-CFP genome by CRISPR-mediated homologous recombination. Specifically, confluent Vero cells in a six-well plate were first infected at MOI = 3 with HSV1-CFP. Three hours later, cells were detached from the plate using Trypsin. Then, 250,000 infected cells were transfected by nucleofection (Lonza, Basel, Switzerland), with 1 µg of the gene fragment and two Cas9 ribonucleoproteins (RNP) specific for the site of integration. The RNPs had a final concentration of 2 µM and were first reconstituted in 3 µL using Cas9 protein, tracrRNA, and crRNA purchased from IDT. We used two crRNA targeting *UL37* to facilitate homologous recombination of the gene fragment (gRNA target sequences: CAATGCACCCAAAGAGCTGC, GGGCGTTCTAAGC CAGACGC). Transfected cells were plated in a single 24-well plate with fresh medium. Twenty-four hours later, serial dilutions of the supernatant containing recombinant viruses were plated into a fresh monolayer of Vero cells and overlaid with 1% methyl-cellulose medium. After 3 days, viral clones were picked, screened by PCR and Sanger sequencing, and clones containing the expected mutation were isolated by three rounds

of serial dilutions and plaque purification. Viral stocks were produced and titered by plaque assay.

McKrae-17 expressing mCitrine fluorescent reporter under a CMV promoter from the *US1/US2* locus was constructed similarly by co-transfecting HSV-1 strain McKrae with a linearized plasmid and a Cas9 RNP targeting the site of integration (*Us1/2* gRNA target sequence: GTCTTAATGGCGGGAAGGG).

## HSV shotgun sequencing

DNA was extracted from viral stocks using Qiagen DNeasy kit. Shotgun libraries were prepared using Illumina DNA Prep kit and sequenced with Illumina Nextseq 2000, generating around 0.5 to 10 million 150 bp paired-end reads. Sequencing results were analyzed using Snippy (17) (https://github.com/tseemann/snippy), using the reference genomes of HSV-1 strain 17+ (Genbank JN555585). Genotypes are provided in Table S1.

## Mouse experiments

All animal procedures were approved by the Institutional Animal Care and Use Committee of the Fred Hutchinson Cancer Center, under protocol numbers 1865. This study was carried out in strict accordance with the recommendations in the Guide for the Care and Use of Laboratory Animals of the National Institutes of Health ("The Guide"). Standard housing, diet, bedding, enrichment, and light/dark cycles were implemented under animal biosafety level 2 (ABSL2) containment. Female Swiss-Webster mice 5 to 6 weeks old were purchased from Charles River Laboratories.

### Ocular infection after corneal scarification

Mice were anesthetized by intraperitoneal injection of ketamine (100 mg/kg) and xylazine (10 mg/kg) and laid under a stereo microscope. Mice corneas were lightly scarified using a 28-gage needle, and 4 µL of viral inoculum was dispensed on both eyes. Following inoculation, ophthalmic drops of local analgesic (Diclofenac) were deposited on both eyes, and the analgesic Meloxicam was added to the drinking water *ad libitum* for 1–5 days following infection. From 5 to 15 days following primary infection, symptoms of infection were reported and scored using an in-house scoring system. Mice experiencing severe symptoms were humanely euthanized.

### Vaginal infection

Mice were treated with 2 mg of Depo-Provera injected subcutaneously. Five to 7 days later, mice were anesthetized by intraperitoneal injection of ketamine (100 mg/kg) and xylazine (10 mg/kg). The vaginal lumen was cleared with a Calginate swab, and 5 µL of viral inoculum was pipetted in the vagina. From 5 to 15 days following primary infection, symptoms of infection were reported and scored using an in-house scoring system. Mice experiencing severe symptoms were humanely euthanized.

### HSV reactivation and quantification of viral loads in swabs and tissues

HSV reactivation was performed by intraperitoneal injection of JQ1 (MedChemExpress, USA) at a dose of 50 mg/kg, as described previously (13–15). DNA extraction from swabs and HSV quantification by qPCR were performed as described previously. Total genomic DNA was isolated from ganglionic tissues using the DNeasy Blood and Tissues kit (Qiagen, Germantown, MD, USA) and eluted in 100 µL of EB buffer, per the manufacturer's protocol. Quantification of the YFP and CFP markers was measured by duplex ddPCR, as described previously (13).

## Statistics and reproducibility

Experiments were carried out in multiple replicates. Investigators were blinded when collecting swabs and analyzing DNA samples. No data were excluded. Statistical analyses

were performed using GraphPad Prism version 10.1.1 for macOS (GraphPad Software, USA, www.graphpad.com).

## ACKNOWLEDGMENTS

We thank members of the Jerome lab for technical and conceptual help.

This study was supported by NIH grant R21AI178255 and through institutional support from the Fred Hutch Cancer Center. In particular, MW received funding from the Evergreen Fund at the Fred Hutch Cancer Center.

M.W. and K.R.J. designed and funded the study. M.W. performed cell culture experiments. M.W., A.K.H., and P.A.M. performed mouse experiments. L.M.K., L.S., and T.K.S. processed samples at the University of Washington Virology Laboratory. H.X. contributed to HSV shotgun sequencing and analysis. M.W. analyzed the data and wrote the manuscript with input from all authors.

## AUTHOR AFFILIATIONS

[1]Vaccine and Infectious Disease Division, Fred Hutch Cancer Center, Seattle, Washington, USA

[2]Department of Laboratory Medicine and Pathology, University of Washington, Seattle, Washington, USA

## AUTHOR ORCIDs

Marius Walter ⓘ http://orcid.org/0000-0002-2476-9661

## FUNDING

| Funder | Grant(s) | Author(s) |
| --- | --- | --- |
| National Institute of Allergy and Infectious Diseases | R21AI178255 | Marius Walter |

## DATA AVAILABILITY

The data supporting the findings of this study are available within the paper and its Supplementary files. Shotgun sequencing data have been deposited in the Short Read Archive with BioProject accession no. PRJNA1302670. Viruses developed in this study are available upon request and subject to standard material transfer agreements with the Fred Hutch Cancer Center.

The code used for the analysis of shotgun sequencing data is described in the method section.

## ADDITIONAL FILES

The following material is available online.

### Supplemental Material

**Data S1 (Spectrum01959-25-s0001.xlsx).** Virus sequencing results.

### Open Peer Review

**PEER REVIEW HISTORY (review-history.pdf).** An accounting of the reviewer comments and feedback.

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
