## [Reviewer comments · Microbiology Spectrum]

Microbiology Spectrum

Herpes simplex virus 1 strain 17+ with R2 mutation in UL37 has residual retrograde transport

Marius Walter, Anoria Haick, Paola Massa, Lindsay Klouser, Laurence Stensland, Tracy Santo, Hong Xie, and Keith Jerome

Corresponding Author(s): Marius Walter, Fred Hutchinson Cancer Center

Review Timeline:

Submission Date:	June 29, 2025
Editorial Decision:	August 14, 2025
Revision Received:	August 15, 2025
Accepted:	August 20, 2025

Editor: Donna Neumann

Reviewer(s): The reviewers have opted to remain anonymous.

Transaction Report:

DOI: <https://doi.org/10.1128/spectrum.01959-25>

Re: Spectrum01959-25 (**Herpes simplex virus 1 strain 17+ with R2 mutation in UL37 has residual retrograde transport**)

Dear Dr. Marius Walter:

Thank you for the privilege of reviewing your work. Below you will find my comments, instructions from the Spectrum editorial office, and the reviewer comments.

I am pleased to inform you that your manuscript has been editorially accepted for publication. However, there are a few additional questions in the submission form that need to be answered before the final decision. Once these are completed, please return your submission so that I can move your paper forward to acceptance.

Revision Guidelines

Sincerely,
Donna Neumann
Editor
Microbiology Spectrum

Reviewer #1 (Comments for the Author):

Walter and colleagues report some standalone observations that expand our risk evaluation on the pathogenesis and potential as vaccines of alphaherpesviruses harboring 5 single-residue point mutations in the so-called R2 domain of the inner tegument

protein UL37. This vaccination concept has been developed by the team of Greg Smith and colleagues, who have shown for HSV-1 strain F, pseudorabiesvirus, and bovine herpesviruses (Pitts et al. 2014, JVI; Richards et al. 2017, PPath; Bernstein et al. 2020, NPJVaccines; Pichard et al. 2020, Vaccine; Stults et al. 2022, JVI). Smith et al. have shown in detailed elegant work that these mutations severely impair retrograde, axonal capsid transport from the periphery towards the neuronal nuclei located in the nervous ganglia.

Walter and colleagues report here that a HSV-1 strain 17+ harboring the equivalent 5 point mutations in UL37 is also severely attenuated in mice upon corneal scarification infection or vaginal infection, and that it also protects against a subsequent infection with a virulent dose of a HSV-1 mosaic isolate of strains 17+ and McKrae. These data confirm that strains with such mutations in the R2 domain of UL37 have the potential to be developed into attenuated vaccine strains.

However, in contrast to the reports by Smith and colleagues on HSV-1 strain F, Walter and colleagues report that their HSV-1 strain 17+ was still able to establish latency in the ganglia of the peripheral nervous system, indicating residual retrograde, axonal transport, and that they could be reactivated, at least after infection via the cornea.

These data contribute to evaluating the efficacy and the risks of such live-attenuated strains of alphaherpesviruses to be developed as vaccine strains.

Specific comments:

1. Abstract: The word "travel" is not used to describe axonal "transport". It would be better to replace travel with transport so that this article shows up in respective data base searches.

2. The wording "contradict published evidence" is too strong (abstract page 1, introduction page 2, page 5, discussion). The authors used the different HSV-1 strain F, and impressively confirmed the strongly reduced pathogenesis of the HSV-1 strain 17+ harboring the same mutations in the R2 domain of UL37. However, the observation that this strain can still establish latency in mice is novel and expands our knowledge on the relevance of the R2 domain for pathogenesis.

3. Importance: The nerves are part of the peripheral nervous system; thus, "... travels through nerves BETWEEN the mucosa and the PNS ..." is a bit misleading and should be reworded.

4. Introduction and discussion: The reference Stults et al. 2022, JVI is missing.

Reviewer #2 (Comments for the Author):

The authors prevent a clear and important observation that the "R2 mutation", previously shown to be completely attenuated in neuron trafficking in a Strain F background, is capable of latently infecting neurons in two separate mouse challenge studies when the mutation is installed in Strain 17+. This observation is important because R2 mutations have been proposed as potential live-attenuated vaccine candidates. Previous studies were performed with the less neurovirulent Strain F, so it makes logical sense to perform the same challenges with highly neurovirulent Strain 17+.

I have no major comments for the authors. If space allows in the observation format, a slightly more in-depth discussion would be appreciated. In particular, I would be interested in knowing if the lack of neuronal trafficking between similar R2 mutations in Strain F and 17+ hold true in different cell models of HSV-1 infection, including human neuronal cells. I would also be interested in more discussion of the seven missense mutations induced in the other viral proteins during creation of the 17+^{R2} strain. While outside the scope of this manuscript, would a true revertant rescue any of these phenotypes?

Rebuttal letter

Dear editors,

Please find below our answer to reviewers' comments following our initial submission to *Microbiology Spectrum*. We are thankful for the overall positive and constructive feedback provided by the reviewers. We have modified the to address the main concerns. The revised manuscript has substantially improved, and we believe that it is now suitable for publication. Please find below a point-by-point answer to the reviewers' comments.

Sincerely,

The authors.

REVIEWER COMMENTS

Reviewer #1 (Comments for the Author):

Walter and colleagues report some standalone observations that expand our risk evaluation on the pathogenesis and potential as vaccines of alphaherpesviruses harboring 5 single-residue point mutations in the so-called R2 domain of the inner tegument protein UL37. This vaccination concept has been developed by the team of Greg Smith and colleagues, who have shown for HSV-1 strain F, pseudorabiesvirus, and bovine herpesviruses (Pitts et al. 2014, JVI; Richards et al. 2017, PPath; Bernstein et al. 2020, NPJVaccines; Pichard et al. 2020, Vaccine; Stults et al. 2022, JVI). Smith et al. have shown in detailed elegant work that these mutations severely impair retrograde, axonal capsid transport from the periphery towards the neuronal nuclei located in the nervous ganglia.

Walter and colleagues report here that a HSV-1 strain 17+ harboring the equivalent 5 point mutations in UL37 is also severely attenuated in mice upon corneal scarification infection or vaginal infection, and that it also protects against a subsequent infection with a virulent dose of a HSV-1 mosaic isolate of strains 17+ and McKrae. These data confirm that strains with such mutations in the R2 domain of UL37 have the potential to be developed into attenuated vaccine strains.

However, in contrast to the reports by Smith and colleagues on HSV-1 strain F, Walter and colleagues report that their HSV-1 strain 17+ was still able to establish latency in the ganglia of the peripheral nervous system, indicating residual retrograde, axonal transport, and that they could be reactivated, at least after infection via the cornea.

These data contribute to evaluating the efficacy and the risks of such live-attenuated strains of alphaherpesviruses to be developed as vaccine strains.

- We thank reviewer 1 for their positive appreciation of our manuscript.
- We agree that it's important to highlight the avirulence and vaccine potential of the R2 mutation confirmed by our study, so a sentence was added to the abstract.

Specific comments:

1. Abstract: The word "travel" is not used to describe axonal "transport". It would be better to replace travel with transport so that this article shows up in respective data base searches.

- We modified the text throughout the manuscript to replace travel by transport.

2. The wording "contradict published evidence" is too strong (abstract page 1, introduction page 2, page 5, discussion). The authors used the different HSV-1 strain F, and impressively confirmed the strongly reduced pathogenesis of the HSV-1 strain 17+ harboring the same mutations in the R2 domain of UL37. However, the observation that this strain can still establish latency in mice is novel and expands our knowledge on the relevance of the R2 domain for pathogenesis.

- In the abstract of "Richards and all, 2017", the author states: "Mutation of this region rendered herpes simplex virus type 1 (HSV-1) and pseudorabies virus (PRV) incapable of spreading by retrograde axonal transport to peripheral ganglia both in culture and animals." Similarly, the titles of "Bernstein et al. 2020, NPJVaccines and Pichard et al. 2020" uses the wording "The R2 non-neuroinvasive HSV-1" and "The pseudorabies virus R2 non-neuroinvasive". In the title and abstracts of these references, the authors do not specify the strain, and their wording suggest that their findings were not strain-specific but applied to all HSV-1 (which was a fair assumption at the time). Thus, we believe that our wording "contradict published evidence" is accurate, since mutation of the UL37 did **not** rendered HSV-1 incapable of retrograde transport.

3. Importance: The nerves are part of the peripheral nervous system; thus, " ... travels through nerves BETWEEN the mucosa and the PNS ..." is a bit misleading and should be reworded.

- We thank the reviewer for this suggestion but believe that knowledgeable reader will be able to make the distinction.

4. Introduction and discussion: The reference Stults et al. 2022, JVI is missing.

- Thank you for providing this reference that we had missed. We added it to the manuscript.

Reviewer #2 (Comments for the Author):

The authors prevent a clear and important observation that the "R2 mutation", previously shown to be completely attenuated in neuron trafficking in a Strain F background, is capable of latently infecting neurons in two separate mouse challenge studies when the mutation is installed in Strain 17+. This observation is important because R2 mutations have been

proposed as potential live-attenuated vaccine candidates. Previous studies were performed with the less neurovirulent Strain F, so it makes logical sense to perform the same challenges with highly neurovirulent Strain 17+.

I have no major comments for the authors. If space allows in the observation format, a slightly more in-depth discussion would be appreciated. In particular, I would be interested in knowing if the lack of neuronal trafficking between similar R2 mutations in Strain F and 17+ hold true in different cell models of HSV-1 infection, including human neuronal cells. I would also be interested in more discussion of the seven missense mutations induced in the other viral proteins during creation of the 17+^{R2} strain. While outside the scope of this manuscript, would a true revertant rescue any of these phenotypes?

- We thank reviewer 2 for their positive appreciation of our manuscript.
- We agree that comparing side by side the R2 mutation in strains F and 17+ would be interesting. We do not have the R2^F virus in our laboratory but would gladly share our new R2¹⁷ virus to researchers that want to pursue this interesting follow-up.
- Regarding the seven missense mutations identified by sequencing. We would like to note that most studies do not sequence their viruses and that such passenger mutations are thus never reported. Furthermore, a revertant that revert the R2 mutation would still carry these passenger mutation (and probably additional one as well). This make obtaining a “true revertant” challenging, since new mutations appear at every new engineering step.
- Importantly, the presence of these mutations does not change the conclusion of our paper. Our main finding is that the defined R2 mutations do not fully ablate retrograde spread. This is independent of any other mutation present in the viral genome.

Re: Spectrum01959-25R1 (**Herpes simplex virus 1 strain 17+ with R2 mutation in UL37 has residual retrograde transport**)

Dear Dr. Marius Walter:

Your manuscript has been accepted, and I am forwarding it to the ASM production staff for publication. Your paper will first be checked to make sure all elements meet the technical requirements. ASM staff will contact you if anything needs to be revised before copyediting and production can begin. Otherwise, you will be notified when your proofs are ready to be viewed.

Sincerely,
Donna Neumann
Editor
Microbiology Spectrum